# Low Fasting Concentrations of Glucagon in Patients with Very Long-Chain Acyl-CoA Dehydrogenase Deficiency

**DOI:** 10.3390/metabo13070780

**Published:** 2023-06-22

**Authors:** Rasmus Stenlid, Hannes Manell, Rikard Seth, Sara Y. Cerenius, Azazul Chowdhury, Camilla Roa Cortés, Isabelle Nyqvist, Thomas Lundqvist, Maria Halldin, Peter Bergsten

**Affiliations:** 1Department of Medical Cell Biology, Uppsala University, SE75123 Uppsala, Swedensara.cerenius@mcb.uu.se (S.Y.C.); azazul.chowdhury@mcb.uu.se (A.C.);; 2Department of Women’s and Children’s Health, Uppsala University, SE75185 Uppsala, Sweden; hannes.manell@kbh.uu.se; 3Department of Women’s and Children’s Health, Karolinska Institute, SE17177 Stockholm, Sweden

**Keywords:** glucagon, insulin, glucagon like peptide-1, very long-chain acyl-CoA dehydrogenase deficiency, medium-chain acyl-CoA dehydrogenase deficiency, carnitine uptake deficiency, inborn errors of metabolism, pediatrics

## Abstract

(1) Background: Deficiencies of mitochondrial fatty acid oxidation (FAO) define a subgroup of inborn errors of metabolism, with medium-chain acyl-CoA dehydrogenase deficiency (MCAD) and very long-chain acyl-CoA dehydrogenase deficiency (VLCAD) being two of the most common. Hypoketotic hypoglycemia is a feared clinical complication and the treatment focuses on avoiding hypoglycemia. In contrast, carnitine uptake deficiency (CUD) is treated as a mild disease without significant effects on FAO. Impaired FAO has experimentally been shown to impair glucagon secretion. Glucagon is an important glucose-mobilizing hormone. If and how glucagon is affected in patients with VLCAD or MCAD remains unknown. (2) Methods: A cross-sectional study was performed with plasma hormone concentrations quantified after four hours of fasting. Patients with VLCAD (*n* = 10), MCAD (*n* = 7) and CUD (*n* = 6) were included. (3) Results: The groups were similar in age, sex, weight, and height. The glucagon and insulin levels were significantly lower in the VLCAD group compared to the CUD group (*p* < 0.05, respectively). The patients with CUD had glucagon concentrations similar to the normative data. No significant differences were seen in GLP-1, glicentin, glucose, amino acids, or NEFAs. (4) Conclusions: Low fasting concentrations of glucagon are present in patients with VLCAD and cannot be explained by altered stimuli in plasma.

## 1. Introduction

Deficiencies of mitochondrial fatty acid oxidation (FAO) define a subgroup of inborn errors of metabolism [1,2]. Approximately 20 different deficiencies in the fatty acid metabolism have been described, with medium-chain acyl-CoA dehydrogenase deficiency (MCAD) (OMIM #201450) and very long-chain acyl-CoA dehydrogenase deficiency (VLCAD) (OMIM #201475) being two of the most common [3,4,5]. In these patients, the symptoms range from non-apparent to rhabdomyolysis, cardiomyopathy, and sudden death, with the latter symptoms being more frequent in VLCAD and uncommon in MCAD [2]. Hypoketotic hypoglycemia is a feared clinical complication [5,6]; therefore, the treatment focuses on avoiding hypoglycemia to avert catabolic states [7]. The underlying mechanisms of hypoglycemia are not yet fully understood. In particular, it is unknown how the glucose-mobilizing hormone glucagon is regulated in VLCAD patients.

Glucagon is derived from the proglucagon peptide and is secreted from the pancreatic α-cells during the fasting state in order to maintain sufficient blood glucose concentrations [8]. Physiologically, glucagon secretion is stimulated by low plasma glucose concentrations, certain amino acids, and non-esterified fatty acids (NEFAs), while glucagon secretion is inhibited by glucagon-like peptide 1 (GLP-1) and insulin [8,9,10,11,12,13]. GLP-1 is, like glucagon, derived from the proglucagon peptide, and is, after being processed by pro-hormone convertase 1/3, secreted alongside other peptides, such as glicentin, from the enteroendocrine L-cells in the small intestine [14].

In isolated human pancreatic islets, NEFAs stimulate glucagon production and secretion [13,15]. This process depends on the mitochondrial metabolism and ATP production through fatty acid β-oxidation, which is mediated via free fatty acid receptor 1-dependent and -independent pathways [13]. The role of the mitochondrial fatty acid β-oxidation in glucagon secretion has further been investigated in carnitine palmitoyl transferase 1 (CPT1) knockout mice and in isolated human pancreatic islets treated with the CPT1 blocker etomoxir. In both cases, glucagon secretion was impaired [16].

The glucagon concentrations have been shown to be altered in children with other metabolic diseases, such as obesity [15,17,18]. Indeed, hyperglucagonemia is particularly pronounced in obese children with high circulating NEFA concentrations [15]. However, it is still not known whether the fasting concentrations of glucagon, GLP-1, or glicentin are affected in patients with MCAD or VLCAD. 

Based on the importance of fatty acid oxidation for glucagon secretion, we hypothesized that genetically impaired mitochondrial fatty acid β-oxidation would cause reduced glucagon secretion. As a consequence, patients with VLCAD deficiency—and to a lesser extent patients with MCAD deficiency—could have low plasma glucagon concentrations, possibly contributing to their tendency to develop hypoglycemia during prolonged fasting.

## 2. Materials and Methods

Study population: Patients with VLCAD (*n* = 10), MCAD (*n* = 7) and carnitine uptake defect (CUD) (OMIM #212140; *n* = 6) were included.

If treated with sufficient supplementation of L-carnitine, CUD is an asymptomatic disease with plasma concentrations of carnitine and acyl-carnitines within the normal range, indicating intact mitochondrial β-oxidation [19]. Recent flux analyses have demonstrated that the mitochondria from patients with CUD function similar to those of healthy controls [20]. Therefore, the individuals with CUD were used as controls in this study.

All individuals were recruited between February and May of 2019 at the Astrid Lindgren Children Hospital, Karolinska University Hospital, Stockholm, Sweden. The inclusion criteria were being 0–18 years of age and diagnosed with VLCAD or MCAD. For patients with CUD, the inclusion criteria were being 0–18 years of age and having normal levels of acyl-carnitine and carnitine in the plasma and no clinical symptoms. Before sampling, the patients all had their normal diet for breakfast in order to reflect their everyday life. Written and oral informed consent forms were obtained from the individuals participating in this study and their legal guardians, respectively. The study was performed in accordance with the Declaration of Helsinki and the local ethics committee approved the study (registration number 2006/005).

Blood sampling: Venous blood was drawn after a four hour fast using a venous catheter, applied after local anesthesia by a mixture of lidocaine and prilocaine (EMLA^®^, AstraZeneca, Cambridge, UK). The blood samples were drawn in an EDTA vacutainer (Becton Dickinson, Franklin Lakes, NJ, USA), immediately placed on ice, and centrifuged at 2500× *g* for 10 min. The plasma was then aliquoted and stored at −80 °C until the analysis. The four hour fasting period was due to the patient’s clinical treatment regimens preventing patients with VLCAD from safely fasting for periods longer than four hours before increasing the risk of a catabolic and dangerous state. 

Biochemical analyses: The concentrations of glucagon in plasma were determined using an enzyme-linked immunosorbent assay (ELISA, Mercodia, Uppsala, Sweden; catalog # 10-1271-01; RRID: AB_2737304). The assay utilizes one C-terminal and one N-terminal antibody to reduce the cross-reactivity with other pro-glucagon-derived peptides. There is no reported cross-reactivity with glicentin or proglucagon (1–16). The lower limit of detection (LOD) of the assay was 1.5 pmol/L. For three individuals with VLCAD, the fasting glucagon concentrations were lower than the LOD. For these subjects, the glucagon concentrations were estimated by extrapolation of the standard curve. The total GLP-1 was determined using a chemiluminescent immunoassay (Mercodia; catalog # 10-1278-01; RRID: AB_2892202), which detects both the intact isoform GLP-1 (7–36) amide as well as the metabolized isoform GLP-1 (9–36) amide. The specificity rates of the total GLP-1 ELISA are reported to be 100% specificity to the GLP-1 (9–36) amide, with 88% cross-reactivity with the GLP-1 (1–36) amide, 93% cross-reactivity with the GLP-1 (7–36) amide, and non-detectable cross reactivity with the glycine-extended GLP-1 (1–37) amide. The ELISA has no cross-reactivity reported for other proglucagon-derived peptides. Insulin (Mercodia; catalog # 10-1113-01; RRID: AB_2877672) and glicentin (Mercodia, Catalog #10-1273-01, RRID: AB_2884906) were both measured using an ELISA. The insulin ELISA has reported cross-reactivity with both proinsulin and c-peptide of lower than 0.01%. The glicentin ELISA has no reported cross-reactivity to other circulating proglucagon-derived gut peptides. For the glicentin analysis, the sample size was *n* = 5 in the CUD group due to the loss of one sample.

The plasma amino acid concentrations were determined using ion-exchange chromatography with spectrophotometric quantification (Biochrome, Cambridge, UK). Due to a technical issue with one of the runs, the sample sizes for the amino acids were CUD *n* = 5, MCAD *n* = 6, and VLCAD *n* = 9.

The total NEFAs in plasma were determined spectrophotometrically according to the manufacturer’s instructions (WAKO NEFA Assay, Fujifilm Wako Diagnostics, Mountain View, CA, USA).

Statistical analyses: A Shapiro–Wilks test was used to determine whether the material followed the Gaussian distribution or not. When the material followed the Gaussian distribution, a one-way ANOVA was used to determine the statistical differences between the three groups. Tukey’s multiple comparisons test was used for differences between multiple groups. When the material did not follow the Gaussian distribution, a Kruskal–Wallis ANOVA was used for statistical differences between the three groups, and for differences between two groups Dunn’s test was used. A ROUT test (Q = 1%) was used to identify statistical outliers. For the glucagon analysis, one outlier was identified and excluded from the CUD group. For the insulin analysis, one outlier was identified and excluded from the VLCAD group and one from the MCAD group. All statistical analyses were performed using GraphPad Prism v9.2 (GraphPad Software Inc., La Jolla, CA, USA). All data are presented as means ± standard deviations, unless otherwise specified. Here, *p* values < 0.05 were considered significant.

## 3. Results

Clinical characteristics: The subjects of the VLCAD, MCAD, and CUD groups were similar in age, weight, height, BMI, BMI standard deviation score (BMI SDS), and sex (Table 1). Notably, the weight values showed high variance in the MCAD group due to one patient having obesity. However, the weight values did not differ significantly between the three groups.

Altered glucagon in children with VLCAD: Lower fasting glucagon concentrations were observed in children with VLCAD (2.5 ± 1.6 pmol/L) compared to children with CUD (4.7 ± 1.6 pmol/L) (Figure 1). The fasting glucagon concentrations of the children with MCAD were at levels between those of the patients with CUD and VLCAD, and not significantly different to the other groups. The glucagon concentrations were measured after 4 h of fasting, which has been reported to be similar to concentrations measured after extended fasting [21].

Unaffected amino acids and lipids: Given the roles of glucose, amino acids, and NEFAs in glucagon secretion [8,9,10,13], we hypothesized that the lower glucagon concentrations in patients with VLCAD could be explained by altered plasma concentrations of glucose, amino acids, or NEFAs. However, the glucose (Figure 2a), total amino acid (Figure 2b), branched-chain amino acid (BCAA), non-BCAA, glucogenic amino acid, all individual amino acid (Table 2), and NEFA (Figure 2c) levels were all of similar plasma concentrations when comparing patients with VLCAD to patients with MCAD or CUD.

Low insulin levels but normal proglucagon hormones in children with VLCAD: Next, we measured the concentrations of insulin and GLP-1, given their inhibitory effects on glucagon secretion [8]. We also measured glicentin, which in addition to GLP-1 is a product of the proglucagon gene [18]. Whereas insulin was significantly lower in patients with VLCAD (2.3 ± 0.9 mIE/L) compared to patients with CUD (4.5 ± 2.9 mIE/L) (Figure 3a), neither GLP-1 nor glicentin differed between the groups (Figure 3b,c).

When including the outliers, the difference in fasting plasma glucagon levels was even greater between CUD patients (7.4 ± 6.5 pmol/L) and VLCAD patients (2.5 ± 1.6 pmol/L) (Figure 4a). When outliers were included in the insulin analysis, no significant differences were seen between the groups (Figure 4b).

## 4. Discussion

This study shows that the fasting glucagon concentrations are lowered in patients with VLCAD, i.e., the genetically-induced impairment of fatty acid β-oxidation appears to be associated with hypoglucagonemia also in humans. The lowered fasting glucagon concentrations seen in patients with VLCAD were not related to the plasma concentrations of glucose, amino acids, or NEFAs. Hence, the results support the notion that impaired fatty acid β-oxidation contributes to lowered glucagon secretion [13,16]. Patients with CUD had fasting glucagon concentrations similar to those reported for healthy controls [15,18,22,23].

The amino acid concentrations were similar in patients with VLCAD, MCAD, and CUD. Amino acids are regarded as major stimulants of glucagon secretion [24]. In a mouse model of VLCAD, Houten et al. showed that gluconeogenesis does not compensate for increased glucose demand, not because of reduced FAO but because of a shortage of glucogenic precursors [25]. However, we could see no differences between the plasma concentrations of individual amino acids, total amino acids, BCAAs, non-BCAAs, or glucogenic amino acids in patients with VLCAD, MCAD, and CUD. It appears that in the patients with VLCAD, amino acids cannot act as satisfactory alternative substrates for mitochondrial metabolism, particularly in glucagon-producing cells.

Since glucagon secretion is also stimulated by NEFAs [15,25], low plasma concentrations of NEFAs could potentially explain the hypoglucagonemia seen in patients with VLCAD. However, we noticed no differences in NEFA concentrations between the three groups, meaning differences in NEFAs cannot explain the observed differences in glucagon. Recently, Briant et al. showed that during hypoglycemia, glucagon secretion by human and mouse pancreatic α-cells is driven by their intrinsic fatty acid β-oxidation [16]. Our results further emphasize the importance of fatty acid β-oxidation in glucagon-producing α-cells to the maintenance of normal plasma glucagon concentrations, at least in fasting.

The plasma glucagon concentrations varied between the VLCAD subjects. In three individuals with VLCAD, the plasma glucagon concentrations were very low, and we speculated that the low glucagon concentrations were caused by comparatively higher concentrations of glucose. This, however, did not turn out to be true. Neither at the group nor individual level was glucagon related to deviations in glucose. This is in line with studies on glucagon receptor knock-out mice, which have shown that an absence of glucagon action does not cause hypoglycemia [9,26]. These results reinforce the notion that during fasting, glucagon secretion is driven by fatty acid β-oxidation [16], which could partly explain how fatty acid β-oxidation deficiencies affect glucagon concentrations in plasma. The glucagon concentrations during fasting were not significantly reduced in patients with MCAD. The patients with MCAD showed less severe symptoms as a group. This observation may reflect the fact that in healthy humans, the majority of the circulating fatty acids are long-chain fatty acids with 16 carbons or more [27]. Hence, the impaired oxidation of long-chain fatty acids, specifically as in VLCAD, may have a greater impact on glucagon production and secretion than impairment of the oxidation of medium-chain fatty acids, as in MCAD [27,28]. However, this should be further investigated.

The low glucagon concentrations in VLCAD subjects could be the result of altered levels of insulin. High concentrations of insulin have been shown to inhibit glucagon secretion via SGLT2-dependent paracrine effects mediated by somatostatin [29]. However, the insulin concentrations at fasting were not higher in the VLCAD group. In fact, the fasting insulin concentrations were lower in the VLCAD group than in the CUD group. This is in line with the fact that insulin secretion in fasting conditions also depends on fatty acid β-oxidation. Indeed, in isolated human pancreatic islets, the pharmaceutical inhibition of the long-chain FAO will markedly decrease both insulin and glucagon secretion [13]. This was confirmed in the human β-cell line EndoC-βH1, where the inhibition of CPT-1 lowered insulin secretion [13]. Thus, the reduced insulin concentrations may reflect the global deficiency of handling nutrients, which is speculated to be present in patients with VLCAD.

Altered GLP-1 concentrations could also potentially cause the low glucagon concentrations in VLCAD subjects, since GLP-1 receptor activation on the pancreatic α-cell inhibits glucagon secretion [30]. However, the GLP-1 concentrations in the VLCAD subjects were not different from the CUD subjects. Since GLP-1 is mainly secreted following a meal [30], and its secretion is possibly driven by glucose metabolism [31], it could be speculated that the GLP-1 concentrations would be lower in children with VLCAD in the fed state, in a similar manner to glucagon and insulin in this study. This remains to be investigated.

Glicentin’s role in human metabolism is not yet fully determined. While some results indicate a role in glucose homeostasis through the augmentation of insulin secretion and inhibition of glucagon secretion [32], other studies have found no effect on insulin secretion [33,34]. However, glicentin is secreted alongside GLP-1 and has been suggested to serve as a marker for L-cell function [35]. We found no differences between the groups, indicating a normal L-cell function during fasting in these patients.

Mitochondrial fatty acid β-oxidation is essential for gluconeogenesis, since it provides energy during fasting [36]. Indeed, hypoglycemia is one of the most severe complications in patients with disorders of mitochondrial fatty acid β-oxidation [37]. To avoid clinical complications, patients need to avoid fasting [2]. In situations where the risk of entering a catabolic state is high, such as infections, prolonged exercise, or surgery, patients are treated with carbohydrates to prevent catabolism [38]. Today, the pharmaceutical alternatives for VLCAD are limited. Medium-chain triacylglycerides (MCT) bypass the enzymes from long-chain fatty acid β-oxidation, and supplementation with MCT fats is helpful for VLCAD patients [7]. Treatment with triheptanoin, a triacylglyceride composed of three seven carbon fatty acids (C7:0), has been associated with reducing the number of major metabolic complications, and is currently under investigation [37,39] and since 2020 has been approved for use in the United States by the Food and Drugs Administration (FDA) [40].

The emergency treatment for a metabolic crisis in FAO today consists of giving the patient large amounts of glucose to prevent cellular lysis, especially rhabdomyolysis [38]. Since the main pharmacological use of glucagon today is as a rescue treatment for severe hypoglycemia, especially in patients with diabetes mellitus [8], patients with VLCAD suffering from an acute metabolic crisis might also benefit from treatment with glucagon. However, the infusion of glucagon into overnight-fasted mice with VLCAD did not affect the blood glucose concentration, which was attributed to depleted glycogen stores and a lack of glucogenic precursors [25]. In addition, the extent to which glucagon stimulates lipolysis and fatty acid oxidation in humans is still unknown [41,42]. Caution needs to be taken if clinically administrating glucagon to patients with FAO deficiencies, due to the possibility of increased lipolysis and fatty acid oxidation, possibly increasing the levels of β-oxidation intermediates [42]. However, in a situation where a patient presents with severe hypoglycemia, the acute risk must be managed immediately, and in this situation the benefits of glucagon treatment could outweigh the risks. This needs to be studied intensely on both a cellular level and in animal models before administration in humans can be tested.

This study is limited by the small number of participants in each group due to the rarity of the disorders and because no healthy control group was included. Hence, this study can be viewed as a pilot study, confirming the preclinical findings regarding glucagon secretion [16]. An arginine stimulation test is considered to be a good way of measuring glucagon secretion [43]; however, this study did not include an arginine stimulation test. The patients with VLCAD had a different clinical severity of the disorder. However, due to the small number of patients, it was not meaningful to relate the clinical severity of the disorder to the glucagon levels. Furthermore, the fasting time was only four hours, mainly due to the character of the disorders, where prolonged fasting could be deleterious. However, in a study by Bieneck et al. on patients with long-chain 3-hydroxyacyl-CoA dehydrogenase deficiency (LCHAD) (OMIM #609016), another β-oxidation defect, it was shown that the lipolysis began to increase already after four hours of fasting [44]. Future studies with a larger number of participants and the addition of a carefully selected healthy control group, postprandial samples, and arginine stimulation test would be of great interest.

## 5. Conclusions

In conclusion, the fasting glucagon concentrations are lowered in patients with VLCAD deficiency, and this is likely driven by the defect of FAO in itself rather than altered factors in the plasma. These results are compatible with α-cell fatty acid β-oxidation being central to appropriate hormonal secretion during fasting.

## Figures and Tables

**Figure 1 metabolites-13-00780-f001:**
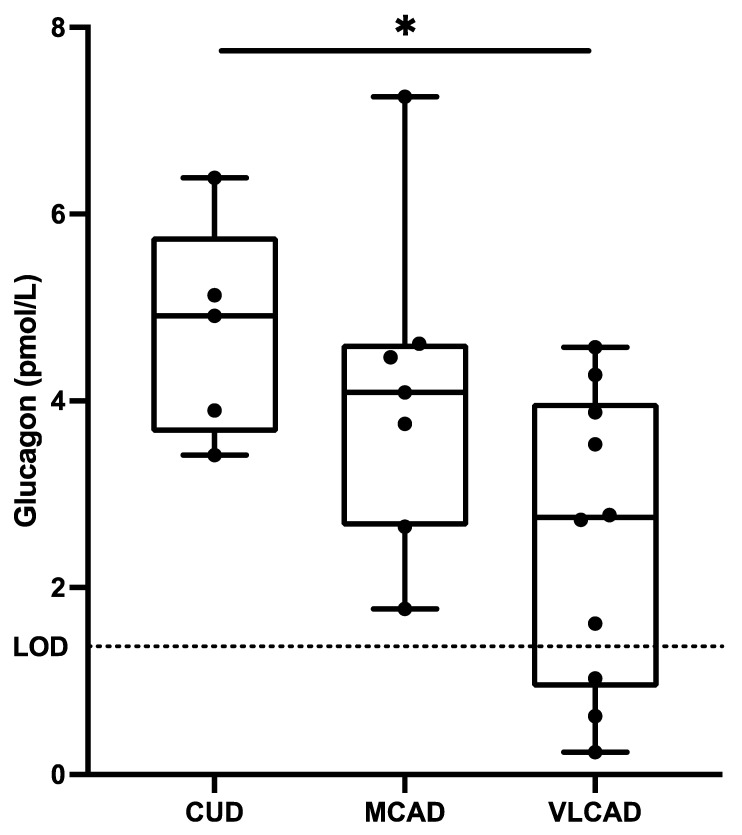
Fasting plasma concentrations of glucagon in patients with carnitine uptake deficiency (CUD, *n* = 5), medium-chain acyl-CoA dehydrogenase deficiency (MCAD, *n* = 7), and very long-chain acyl-CoA dehydrogenase deficiency (VLCAD, *n* = 10). The lower limit of detection (LOD) was 1.5 pmol/L. Each black point represents an individual patient. Note: * = *p* < 0.05.

**Figure 2 metabolites-13-00780-f002:**
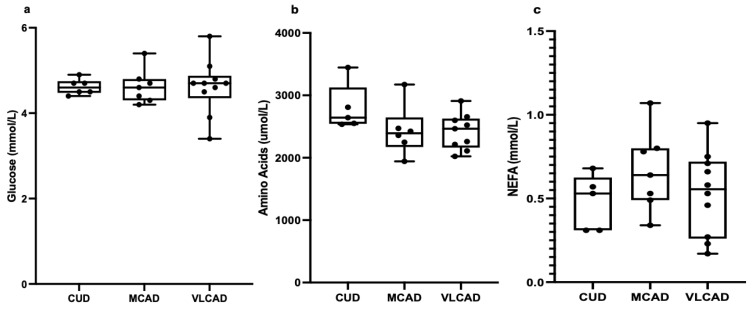
Fasting plasma concentrations of glucose (**a**), amino acids (**b**), and non-esterified fatty acids (NEFA) (**c**) in patients with carnitine uptake deficiency (CUD, *n* = 6), medium-chain acyl-CoA dehydrogenase deficiency (MCAD, *n* = 7), and very long-chain acyl-CoA dehydrogenase deficiency (VLCAD, *n* = 10). For the amino acids, *n* = −1 for each group. Each black point represents an individual patient.

**Figure 3 metabolites-13-00780-f003:**
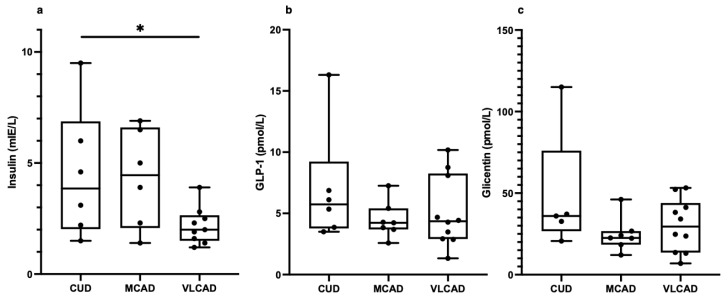
Fasting plasma concentrations of insulin (**a**), glucagon-like peptide 1 (GLP-1) (**b**), and glicentin (**c**) in patients with carnitine uptake deficiency (CUD, *n* = 6), medium-chain acyl-CoA dehydrogenase deficiency (MCAD, *n* = 7), and very long-chain acyl-CoA dehydrogenase deficiency (VLCAD, *n* = 10). For insulin, *n* = −1 in the MCAD and VLCAD group. For glicentin, *n* = −1 in the CUD group. Each black point represents an individual patient. Note: * = *p* < 0.05.

**Figure 4 metabolites-13-00780-f004:**
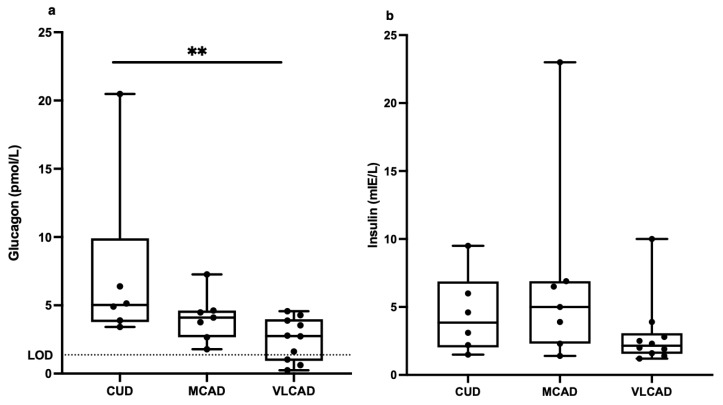
Fasting plasma concentrations of glucagon (**a**) and insulin (**b**) in patients with carnitine uptake deficiency (CUD, *n* = 6), medium-chain acyl-CoA dehydrogenase deficiency (MCAD, *n* = 7), and very long-chain acyl-CoA dehydrogenase deficiency (VLCAD, *n* = 10). Statistical outliers included. The lower limit of detection (LOD) for glucagon was 1.5 pmol/L. Each black point represents an individual patient. Note: ** = *p* < 0.01.

**Table 1 metabolites-13-00780-t001:** Clinical characteristics of the study group. Presented as means ± SDs. CUD = carnitine uptake deficiency; MCAD = medium-chain acyl-CoA dehydrogenase deficiency; VLCAD = very long-chain acyl-CoA dehydrogenase deficiency.

	CUD (*n* = 6)	MCAD (*n* = 7)	VLCAD (*n* = 10)	*p* Value
Age, y	6.8 ± 3.2	7.6 ± 3.5	5.25 ± 2.58	0.29
Weight, kg	27.0 ± 14.0	36.1 ± 31.6	22.5 ± 8.0	0.41
Height, cm	117.0 ± 24.2	126.7 ± 20.6	113.3 ± 14.3	0.39
BMI	17.8 ± 4.0	21.1 ± 8.6	17.1 ± 2.6	0.37
BMI SDS	0.70 ± 1.21	1.27 ± 1.6	0.64 ± 1.22	0.64
Sex (female/male)	2/4	3/4	3/7	0.87

**Table 2 metabolites-13-00780-t002:** Amino acids. All values presented as means ± standard deviations (SDs) in µmol/L. AA = amino acids; BCAA = branched-chain amino acids; CUD = carnitine uptake deficiency; MCAD = medium-chain acyl-CoA dehydrogenase deficiency; VLCAD = very long-chain acyl-CoA dehydrogenase deficiency.

	CUD (*n* = 5)	MCAD (*n* = 6)	VLCAD (*n* = 9)	*p* Value
BCAA	398 ± 93	323 ± 78	362 ± 66	0.68
Non-BCAA	2399 ± 297	2113 ± 330	2056 ± 245	0.99
Glucogenic AA	1875 ± 173	1698 ± 294	1667 ± 201	0.27
α-Aminobutyric acid	22 ± 9	25 ± 11	19 ± 6	0.99
Alanine	322 ± 66	289 ± 103	251 ± 55	0.70
Arginine	79 ± 28	63 ± 17	62 ± 21	0.99
Asparagine	46 ± 3	44 ± 8	44 ± 9	0.99
Aspartic acid	15 ± 3	16 ± 4	12 ± 1	0.99
Citrulline	34 ± 4	36 ± 5	40 ± 9	0.99
Cysteine	<4	4 ± 3	5 ± 4	0.95
Glutamic acid	118 ± 39	76 ± 12	67 ± 12	0.98
Glutamine	473 ± 60	475 ± 50	436 ± 66	0.67
Glycine	211 ± 33	177 ± 36	216 ± 29	0.66
Histidine	73 ± 5	62 ± 6	64 ± 8	0.99
Isoleucine	66 ± 24	45 ± 11	53 ± 16	0.98
Leucine	123 ± 29	93 ± 21	106 ± 22	0.95
Lysine	157 ± 54	127 ± 27	131 ± 36	0.99
Methionine	23 ± 6	20 ± 5	22 ± 6	0.99
Ornithine	50 ± 20	54 ± 20	43 ± 10	0.97
Phenylalanine	83 ± 65	58 ± 10	48 ± 6	0.97
Proline	192 ± 33	194 ± 93	180 ± 51	0.95
Serine	113 ± 28	98 ± 13	110 ± 31	0.96
Taurine	149 ± 69	110 ± 27	88 ± 32	0.89
Threonine	107 ± 22	91 ± 24	118 ± 45	0.83
Tryptophan	44 ± 9	30 ± 8	38 ± 11	0.98
Tyrosine	86 ± 26	71 ± 16	64 ± 13	0.99
Valine	209 ± 41	186 ± 47	203 ± 35	0.92
Alloisoleucine	<4	<4	5 ± 2	0.99

## Data Availability

The data presented in this study are available on request from the corresponding author. The data are not publicly available due to being clinical data.

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
