# Peer review of "Low Fasting Concentrations of Glucagon in Patients with Very Long-Chain Acyl-CoA Dehydrogenase Deficiency"

_metabolites, 2023, doi:10.3390/metabo13070780_

Round 1

Reviewer 1 Report

This study offers valuable insights into the metabolic implications of VLCAD deficiency and its association with fasting glucagon levels.

You conducted a well-designed study, employing rigorous methodologies and comprehensive data analysis. The clarity of the writing and the logical organization of the article makes it highly accessible and engaging to readers from diverse scientific backgrounds.

The findings presented in this article are significant and contribute to our understanding of VLCAD deficiency.

Author Response

We are extremely grateful for these wonderful comments.  

Reviewer 2 Report

The authors analyzed glucagon concentration in patients with a deficiency in FAO. They found a low level of glucagon and insulin in patients with VLCAD; the altered stimuli in plasma could not explain this. However, the study has some limitations, the most important being the low number of patients included and the lack of control (as we discuss rare diseases).

Some minor improvements may be made: The paper's abstract must be rewritten to include more data and results. In addition, the abbreviated words should not be listed in the keywords.

The manuscript is generally well written: the Introduction explains well the importance of the study; the Methods are correctly presented; the Results support the Study's Conclusions. The Discussions put the results in the proper analysis of their importance for the deficiencies in FAO patients. The authors mentioned in a fair way the limitations of the study and the need to be considered as a pilot study keeping in mind the low number of subjects.

Some minor improvements should be made to the use of the English language. 

Author Response

Point 1: The paper's abstract must be rewritten to include more data and results. In addition, the abbreviated words should not be listed in the keywords.

Response 1: Thank you for this suggestion. We have rewritten the abstract and emphasized on expanding the results section. In addition, we have removed the abbreviated words as keywords and instead written out the full terms accordingly.  

Point 2: Some minor improvements should be made to the use of the English language. 

Response 2: We have read through the manuscript with a language editor and improved the English language according to the suggestions.

Thank you very much for you contribution to improving the manuscript.

Sincerely,

The authors

Reviewer 3 Report

The paper is well written and deals with a very interesting topic.

I only suggest minor revisions.

The abstract draws the reader's attention to glucagon and insulin, while the discussion is entirely focused on glucagon. I suggest correcting this discrepancy, either in the abstract or in the discussion

Line 46: what does PP mean?

Lines 127-129: you say that age, sex, weight and height did not differ significantly between the three groups. However, the p-value is not shown in table 1. Please enter it

Table 2: as above: enter the p-value

Lines 262-264: I don’t understand this sentence: “This patient group” refers to MCAD patients? In MCAD patients “the majority of the circulating  fatty acids are long-chain fatty acids with 16 carbons or more”?

Author Response

Response to Reviewer 3 Comments

Point 1: The abstract draws the reader's attention to glucagon and insulin, while the discussion is entirely focused on glucagon. I suggest correcting this discrepancy, either in the abstract or in the discussion

Response 1: Thank you for this suggestion. We have corrected this discrepancy by editing the abstract. We feel that it improves the readability of the whole paper, and we therefore are very grateful for your suggestion.

Point 2: Line 46: what does PP mean?

Response 2: We have written out the abbreviation.

Point 3: Lines 127-129: you say that age, sex, weight and height did not differ significantly between the three groups. However, the p-value is not shown in table 1. Please enter it.

Response 3: We have entered the p-values to table 1.

Point 4: Table 2: as above: enter the p-value.

Response 4: We have entered the p-values to table 2.

Point 5a: Lines 262-264: I don’t understand this sentence: “This patient group” refers to MCAD patients?

Point 5b: In MCAD patients “the majority of the circulating  fatty acids are long-chain fatty acids with 16 carbons or more”?

Response 5a: Yes, we have clarified the sentence on lines 282- 288 “The patients with MCAD show less severe symptoms as a group. This observation may reflect the fact that in healthy humans, the majority of the circulating fatty acids are long-chain fatty acids with 16 carbons or more [27]. Hence, impaired oxidation of long chain fatty acids, specifically - as in VLCAD, may have a greater impact on glucagon production and secretion than impairment of the oxidation of medium chain fatty acids, as in MCAD [27,28]. However, this should be further investigated.”

Response 5b: No, we have clarified the sentence. Kindly see above.

Thank you very much for you contribution to improving the manuscript.

Sincerely,

The authors
